# Learning to teach: Improving Mean Teacher in Semi-supervised Medical Image Segmentation with Dynamic Decay Modulation

## Abstract

Medical image segmentation is essential in medical diagnostics but is hindered by the scarcity of labeled three-dimensional imaging data, which requires costly expert annotations. Semi-supervised learning (SSL) addresses this limitation by utilizing large amounts of unlabeled data alongside limited labeled samples. The Mean Teacher model, a prominent SSL method, enhances performance by employing an Exponential Moving Average (EMA) of the student model to form a teacher model, where the EMA decay coefficient is critical. However, using a fixed coefficient fails to adapt to the evolving training dynamics, potentially restricting the model's effectiveness. In this paper, we propose Meta Mean Teacher, a novel framework that integrates meta-learning to dynamically adjust the EMA decay coefficient during training. We incorporate the proposed Dynamic Decay Modulation (DDM) module into our Meta Mean Teacher framework, which captures the representational capacities of both student and teacher models. DDM heuristically learns the optimal EMA decay coefficient by taking the losses of the student and teacher networks as inputs and updating it through pseudo-gradient descent on a meta-objective. This dynamic adjustment allows the teacher model to more effectively guide the student as training progresses. Experiments on two datasets with different modalities, i.e., CT and MRI, show that Meta Mean Teacher consistently outperforms traditional Mean Teacher methods with fixed EMA coefficients. Furthermore, integrating Meta Mean Teacher into state-of-the-art frameworks like UA-MT, AD-MT, and PMT leads to significant performance enhancements, achieving new state-of-the-art results in semi-supervised medical image segmentation. We will release the code if the paper is accepted.

## 1 Introduction

Medical image segmentation, as a fundamental task in the medical field, lays the groundwork for subsequent high-level tasks. However, three-dimensional medical imaging data pose a significant annotation challenge, typically requiring the supervision of domain experts for accurate annotations, making it difficult to provide large amounts of data. Semi-supervised learning aims to better utilize extensive unlabeled data under the supervision of limited labeled data (Zhou et al., 2021; Zheng et al., 2022; Rizve et al., 2022; Xia et al., 2023; Xin et al., 2019), effectively addressing limited data availability by fully leveraging unlabeled samples.

Mean Teacher (Tarvainen & Valpola, 2017) is a classic work in the field of semi-supervised learning and can significantly enhance performance in semi-supervised medical image segmentation. Early work in the field, such as UA-MT, improved the Mean Teacher paradigm by introducing uncertainty, thereby enhancing the model's representation capabilities. More recent work, such as PMT (Gao et al., 2024) and AD-MT (Zhao et al., 2023), demonstrated that using a more complex temporal framework, Mean Teacher can still improve the model's representation capabilities.

The Mean Teacher model employs Exponential Moving Average (EMA) to generate a teacher model from the student model without additional training, where the decay coefficient $\alpha$ plays a critical role. As illustrated in Figure 1, varying values of $\alpha$ have a significant impact on model performance. To better harness the potential of the Mean Teacher framework, we propose Meta Mean Teacher,

a framework that utilizes meta-learning and incorporates plug-and-play modules Dynamic Decay Modulation (DDM) for optimization. By introducing DDM, we capture the training representations of the model and explicitly adjust the $\alpha$ value of the Mean Teacher throughout the iterative process.

This framework design is inspired by observations on the influence of EMA decay coefficient in the Mean Teacher method and insights from meta-learning in hyperparameter optimization (Finn et al., 2017; Li et al., 2017; Shu et al., 2019). One interpretation is that $\alpha$ determines the update intensity of the teacher model, a characteristic that should evolve as the model training progresses, which DDM can capture effectively. Specifically, for the DDM module that allows dynamic adjustment of $\alpha$, we aim to enable the module to fully perceive the representational capacity of both student and teacher models and to heuristically explore and learn the optimal $\alpha$ during training. Consistent with previous approaches, we use the losses of both student and teacher models as inputs to the module. Our DDM is trained on meta data unseen by the Mean Teacher architecture and is updated through pseudo-gradient descent.

Building on this foundation, we discuss the design of DDM, specifically focusing on the choice of student guidance as the model update mechanism. Unlike single-model meta-learning, the Mean Teacher framework introduces an interdependent relationship between the teacher and student models. The student leverages pseudo-labels generated by the teacher to fully utilize unsupervised data, while the teacher is updated through the momentum of the student model. DDM requires an appropriate loss function to correctly update its parameters. We chose the loss of the student network as the guiding mechanism for updating DDM, which we argue is superior to the alternative approach of teacher guidance, where the DDM is updated based on the teacher model's loss. The DDM module is obtained through several steps in the meta-phase, which is elaborated on in the method section. In the Experiments section, we also compare the performance of using the teacher loss updated by meta $\alpha$. Furthermore, we provide a simple derivation demonstrating that this loss is more aligned with our objectives.

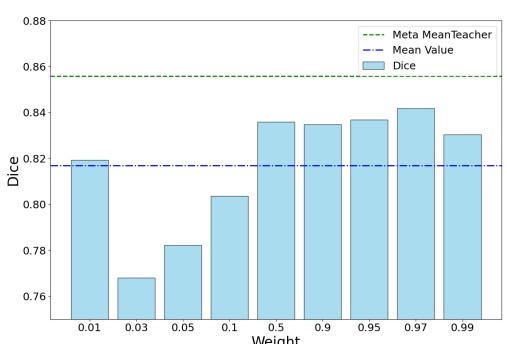

Figure 1: Performance comparison of Mean Teacher with different $\alpha$ values and our proposed DDM on the LA Heart dataset.

The Meta Mean Teacher framework is designed as a pluggable module, enabling deployment in more extensive and advanced models. Our experiments show that Meta Mean Teacher outperforms traditional Mean Teacher methods with a fixed EMA coefficient. Moreover, integrating Meta Mean Teacher into UA-MT, AD-MT and PMT frameworks also leads to performance improvements, while the latter two are the current SOTA methods.

The main contributions of this paper are as follows:

1. We propose the Meta Mean Teacher training framework, which can serve as a pluggable module to enhance the performance of methods employing the Mean Teacher approach, demonstrating its versatility across different Mean Teacher-based methods.

2. We introduce DDM, which dynamically generates the EMA decay coefficient based on the losses of both teacher and student within the Mean Teacher framework.

3. Our experimental results indicate performance improvements across methods using the Mean Teacher architecture, with AD-MT experiments achieving new SOTA performance.

## 2 RELATED WORKS

### 2.1 SEMI-SUPERVISED LEARNING

Semi-supervised learning leverages both labeled and unlabeled data to enhance model performance, primarily through two paradigms: consistency regularization and pseudo label generation. Consistency regularization aims to maintain stable predictions under various perturbations. For instance,

the Π model (Laine & Aila, 2016) applies image-level perturbations, while the Mean Teacher (MT) model (Tarvainen & Valpola, 2017) uses exponential moving average (EMA) to align outputs between teacher and student models. Advanced methods like SASSnet (Li et al., 2020) focus on geometric shape regularity, and CPCL (Xu et al., 2022) establishes a cyclic framework for supervised and unsupervised training regularization. Further, DTC (Luo et al., 2021) introduces task-level regularization with a dual-task consistency framework, and MCF (Wang et al., 2023) employs heterogeneous models for model-level regularization. Pseudo label generation enhances the discriminative ability of models by using labeled data to train a prior model, which then generates pseudo labels for unlabeled data. Direct methods, such as the approach in (Lee et al., 2013), use high-confidence thresholds, while UA-MT (Yu et al., 2019) filters unreliable pseudo labels using uncertainty estimation. Indirect methods include Tri-Net (Dong-DongChen & WeiGao, 2018), which uses two subnetworks to generate pseudo labels for a third, and MCF (Wang et al., 2023), which dynamically generates pseudo labels with a heterogeneous network. These approaches collectively advance semi-supervised learning by enhancing model stability and discriminative capacity through innovative regularization and pseudo labeling techniques.

## 2.2 Meta Learning

Meta-learning or "learning to learn" fundamentally involves optimizing hyperparameters to enhance model adaptability across diverse tasks. This approach aims to create models that can quickly adapt to new tasks with minimal data and computational resources. Several key works have approached this from different angles, each contributing unique methodologies and insights to the field. One of the foundational works in this area is Model-Agnostic Meta-Learning (MAML) (Finn et al., 2017), which optimizes initial model parameters to enable rapid adaptation with minimal data through a few gradient updates. MAML's core idea is to find a set of initial parameters that can be fine-tuned quickly for any given task, making it highly versatile for various applications. Meta-SGD (Li et al., 2017) extends this concept by learning not only the initialization but also the update direction and learning rates. This allows for efficient single-step adaptation, further reducing the computational burden and improving the speed of adaptation. Meta-Weight-Net (Shu et al., 2019) addresses biased training data by learning a weighting function guided by unbiased meta-data, adjusting the importance of training samples to improve generalization. Other notable works also contribute to hyperparameter optimization through meta-learning. Learning to Optimize (Li & Malik, 2016) automates optimization algorithm design using reinforcement learning. MetaQNN (Baker et al., 2016) automates CNN architecture design through Q-learning. Prototypical Networks (Snell et al., 2017) learn a metric space for classification based on class prototypes. Meta Networks (MetaNet) (Munkhdalai & Yu, 2017) adjust inductive biases for rapid generalization. ALFA (Baik et al., 2020) enhances MAML by adaptively generating hyperparameters like learning rates and weight decay coefficients.

## 3 Method

### 3.1 Mean Teacher

Methods like Mean Teacher for semi-supervised datasets involve a set of homomorphic Teacher and Student models, denoted as $f_{\theta^t}(\cdot)$ and $f_{\theta^s}(\cdot)$. In a semi-supervised process, the training data includes a small number of labeled data denoted as $\mathbf{D}_L = \{(x_i^L, y_i^L)\}_{i=1}^N$, and a large amount of unlabeled data denoted as $\mathbf{D}_U = \{(x_i^U)\}_{i=N+1}^{N+M}$, where $N \ll M$, $x_i \in \mathbb{R}^{H \times W \times D}$ represents medical volumes, and $y_i \in \{0, 1\}^{H \times W \times D}$ represents ground truth labels. Batches of input data $\mathbf{X}$ consist of an equal proportion of labeled data $(\mathbf{X}^L, \mathbf{Y}^L)$ and unlabeled data $\mathbf{X}^U$. These volumes are fed into $f_{\theta^t}(\cdot)$ and $f_{\theta^s}(\cdot)$, and the output of the Teacher model is used to construct pseudo-labels and semi-supervised loss: $\hat{y}_i^U = f_{\theta^t}(x_i^U), \quad \forall x_i^U \in \mathbf{D}_U$.

The total loss $\mathcal{L}$ is composed of two parts: the supervised loss $\mathcal{L}_s$ on the labeled data and the unsupervised loss $\mathcal{L}_u$ on the unlabeled data. The supervised loss $\mathcal{L}_s$ is typically the cross-entropy loss between the predictions of the Student model $f_{\theta^s}$ and the ground truth labels $\mathbf{Y}^L$, while the unsupervised loss $\mathcal{L}_u$ is the consistency loss between the predictions of the Student model $f_{\theta^s}$ and the pseudo-labels $\hat{y}_i^U$ generated by the Teacher model:

$$\mathcal{L}_s = \frac{1}{N}\sum_{i=1}^{N}\mathcal{L}_{ce}(f_{\theta^s}(x_i^L), y_i^L), \quad \mathcal{L}_u = \frac{1}{M}\sum_{i=N+1}^{N+M}\mathcal{L}_{cons}(f_{\theta^s}(x_i^U), \hat{y}_i^U),$$

where $\mathcal{L}_{cons}$ can be a MSE loss or other suitable consistency loss function.

The total loss is then a weighted sum of the supervised and unsupervised losses, $\mathcal{L} = \mathcal{L}_s + \lambda\mathcal{L}_u$, where $\lambda$ is a weight factor that balances the contribution of the supervised and unsupervised losses. During training, the parameters of the Student model $\theta^s$ are updated to minimize the total loss $\mathcal{L}$, while the Teacher model parameters $\theta^t$ are updated as an EMA of the Student model parameters:

$$\theta_{t+1}^t \leftarrow \alpha\theta_t^t + (1-\alpha)\theta_{t+1}^s.$$

This Mean Teacher architecture leverages a more stable Teacher model to utilize a large amount of unsupervised data during training and avoids overfitting on the limited supervised data. However, the key parameter in this process, the EMA decay coefficient $\alpha$, is seldom discussed, which often leads to the Mean Teacher framework not being fully utilized to its potential.

For the key parameter $\alpha$ in the Mean Teacher framework, a naive understanding is that if $\alpha$ is too large, the Teacher model will be too conservative and difficult to absorb new knowledge learned by the Student model; if $\alpha$ is too small, the Teacher model may become too sensitive and easily affected by noise. Figure 1 shows the significant impact of $\alpha$ on the performance of the Mean Teacher. On the other hand, for the LA Heart dataset, compared to the common value of $\alpha$ being 0.99, a more appropriate value such as 0.97 can bring an improvement of nearly 1 in dice loss, while an intuitively small value of 0.01 does not cause a significant performance degradation.

## 3.2 META MEAN TEACHER FRAMEWORK

Our Meta Mean Teacher framework consists of two main components: Mean Teacher and DDM module. Figure 2 illustrates the overall pipeline of our method.

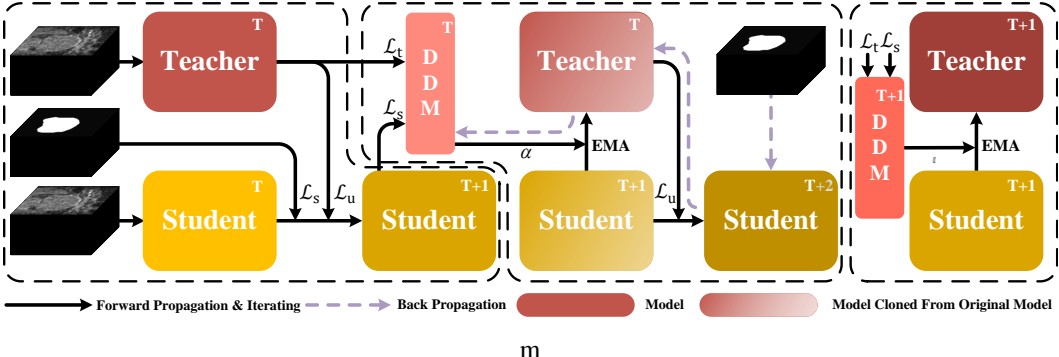

m

Figure 2: The pipeline of Meta Mean Teacher. The left side shows student-learning phase, the middle side shows meta-learning phase, and the right side shows teacher-updating phase using DDM.

As shown in Figure 2, our method operates in two phases: the meta-learning phase and the normal training phase. In the meta-learning phase, we use a student-guided approach to update the DDM module. In the normal training phase, we use the updated DDM to generate dynamic $\alpha$ values for EMA updates of the teacher model.

The DDM module is the core component of our Meta Mean Teacher framework. It is designed to generate dynamic $\alpha$ values based on the current state of both the student and teacher models. The DDM is implemented as a small neural network, typically a MLP, that takes as input the losses of the student and teacher models on the labeled data.

Through pseudo-gradient descent, the loss function can be passed through the optimizer, thereby establishing a computational graph from the loss function to the DDM in most cases. However,

this raises another question: how to choose this loss function to guide the update of the DDM. One approach is to directly compute the loss of the Teacher model updated using the DDM, which we refer to as the teacher-guided meta-learning approach. In contrast, we use a relatively complex method, the student-guided meta-learning approach, where the loss is guided by the student being supervised and iterated by the Teacher updated using the DDM. Our student-guided meta-learning approach consists of the following steps:

---

**Algorithm 1** Student-Guided Meta-Learning

---

1: **Input:** Labeled data $\mathcal{D}_s$, Unlabeled data $\mathcal{D}_u$, Meta data $\mathcal{D}_m$
2: **Output:** Updated student model $\theta^s$, teacher model $\theta^t$, and DDM parameters $\theta^D$
3: **for** each training iteration **do**
4:     **Student-Learning Phase:**
5:     Compute student loss: $\mathcal{L}_s(\theta_t^s, \mathcal{D}_s)$
6:     Update student model: $\theta_{t+1}^s = \theta_t^s - \eta \nabla_{\theta^s} \mathcal{L}_s(\theta_t^s, \mathcal{D}_s)$
7:     **Meta-Learning Phase:**
8:     Clone student and teacher model parameters: $\theta_{\text{clone}}^s \leftarrow \theta_{t+1}^s$, $\theta_{\text{clone}}^t \leftarrow \theta_t^t$
9:     **for** each meta iteration **do**
10:         Compute DDM output: $\alpha_m = f^D(\mathcal{L}_s(\theta_{\text{clone}}^s, \mathcal{D}_s), \mathcal{L}_s(\theta_{\text{clone}}^t, \mathcal{D}_s))$
11:         Update cloned teacher model: $\theta_{\text{clone}}^t = \alpha_m \theta_{\text{clone}}^t + (1 - \alpha_m)\theta_{\text{clone}}^s$
12:         Generate pseudo-labels: $\hat{y}_u = f_t(\theta_{\text{clone}}^t, x_u), \forall x_u \in \mathcal{D}_u$
13:         Compute total loss: $\mathcal{L} = \mathcal{L}_s(\theta_{\text{clone}}^s, \mathcal{D}_s) + \lambda \mathcal{L}_u(\theta_{\text{clone}}^s, \mathcal{D}_u, \hat{y}_u)$
14:         Update cloned student model: $\theta_{\text{clone}}^s = \theta_{\text{clone}}^s - \eta \nabla_{\theta^s} \mathcal{L}$
15:         Compute meta loss: $\mathcal{L}_D = \mathcal{L}_m(\theta_{\text{clone}}^s, \mathcal{D}_m)$
16:         Update DDM: $\theta^D = \theta^D - \eta_D \nabla_{\theta^D} \mathcal{L}_D$
17:     **end for**
18:     **Teacher-updating Phase**
19:     Compute DDM output: $\alpha_m = f_D(\mathcal{L}_s(\theta_{t+1}^s, \mathcal{D}_s), \mathcal{L}_s(\theta_t^t, \mathcal{D}_s))$
20:     Update teacher model: $\theta_{t+1}^t = \alpha_m \theta_t^t + (1 - \alpha_m)\theta_{t+1}^s$
21: **end for**

---

In this algorithm, $\mathcal{L}_u$ represents the unsupervised loss function, $\lambda$ is a weight balancing the supervised and unsupervised losses. Besides, $\mathcal{L}_m$ is the loss function on the meta dataset, which takes the same form as $\mathcal{L}_s$ and transforms the parameter optimization of DDM into a bilevel optimization problem:

$$\Theta_D^* = \arg\min_{\Theta_D} \mathcal{L}_m(\Theta_s^* \triangleq \arg\min_{\Theta_s} \mathcal{L}(\Theta_s, \Theta_t, \mathcal{D}_{s\cup u}), \mathcal{D}_m), \tag{1}$$

where $\Theta_t$ is computed with EMA using $\alpha_m$, $\mathcal{D}_s$ is the meta data, and $\alpha_m$ is the output of DDM:

$$\alpha_m = f_D(\mathcal{L}_s(\theta_t^s, \mathcal{D}_s), \mathcal{L}_s(\theta_t^t, \mathcal{D}_s)), \tag{2}$$

where $\mathcal{L}_s$ is the supervised loss function, $\theta_t^s$ and $\theta_t^t$ are the parameters of the student and teacher models at time step $t$, respectively, and $\mathcal{D}_s$ is the labeled data.

To understand why our student-guided approach is more effective than a teacher-guided one, we provide a theoretical analysis based on function approximation in parameter space.

Let $\mathcal{F}$ be the space of all possible medical image segmentation functions, and $p^* \in \mathcal{F}$ be the true segmentation function. The student model can be represented as $f_s : \Theta_s \to \mathcal{F}$, where $\Theta_s$ is the parameter space of the student model. Similarly, the teacher model can be represented as $f_t : \Theta_t \to \mathcal{F}$.

Essentially, in the teacher-guided approach, the iterative process aims to have the teacher learn the optimal interpolation result between the student and the previous teacher parameters. However, the teacher can provide an unbiased estimate of the medical image segmentation only if the line segment between the previous teacher and student parameters passes through the optimal parameters. In a high-dimensional parameter space, this probability approaches zero.

$$\mathbb{P}\left(\theta^* \in \text{span}(\theta_t^{(k)}, \theta_s^{(k)})\right) \approx 0 \quad \text{for large dimensions.} \tag{3}$$

The EMA update defines a mapping $g : \Theta_t \times \Theta_s \to \Theta_t$. Due to the linear nature of $g$, there exists an unavoidable approximation error:

$$\exists \epsilon > 0, \forall \theta^t \in \Theta_t, \theta^s \in \Theta_s, \|f_t(g(\theta^t, \theta^s)) - p^*\|_{\mathcal{F}} > \epsilon. \tag{4}$$

Regardless of the choice of student and teacher model parameters, the teacher model after EMA updates will always be at a certain distance from the true segmentation function. Specifically, if we understand the EMA process as performing linear interpolation in the parameter space, learning the EMA parameter $\alpha$ essentially attempts to find a linear interpolation between $\Theta_s$ and $\Theta_t$ that approximates the minimum value along the line segment connecting them. In reality, this minimum value might lie at a saddle point, which can be far from $p^*$.

In contrast, the student model is not constrained by EMA updates. According to the Universal Approximation Theorem, given a sufficiently complex network structure, the student model can theoretically approximate the true segmentation function to arbitrary precision:

$$\forall \epsilon > 0, \exists \theta^s \in \Theta_s, \|f_s(\theta^s) - p^*\|_{\mathcal{F}} < \epsilon. \tag{5}$$

By updating the EMA based on the student model, the teacher model after EMA updates tends to provide pseudo-labels that better help the student model approximate $p^*$, thereby enhancing the performance of the student model.

This theoretical analysis provides crucial insights into the effectiveness of our student-guided approach. By using the student model to guide the learning process, we shift the focus from attempting to create an optimal teacher model (which is inherently limited by EMA constraints) to creating a teacher model that best facilitates the student's learning process.

In other words, the DDM module in our approach doesn't learn to produce a teacher model that directly approximates the true distribution—a goal that is theoretically unattainable due to EMA constraints. Instead, it learns to generate a teacher model that, while not necessarily optimal in isolation, is ideally suited to guide the student model towards the true distribution through pseudo-labeling and other interactions.

In essence, our method transforms the role of the teacher model from a target to be reached into a guide to be followed. This paradigm shift allows us to overcome the theoretical limitations of EMA-based approaches while still benefiting from the stability and regularization effects that make Mean Teacher methods effective in semi-supervised learning scenarios.

## 4 EXPERIMENTS

### 4.1 IMPLEMENTATION DETAILS

We selected the VNet (Milletari et al., 2016) model as a baseline network, which performs well in conditions with limited data and is essentially a 3D convolutional version of UNet. During inference, we use the average of the outputs from two networks as the final prediction. Specifically, the SGD optimizer was used to update the network parameters with weight decay of $0.0001$ and a momentum of $0.9$. The initial learning rate was set to $0.01$, reduced by a factor of 10 every 2500 iterations, for a total of 6000 iterations.

Following the practice in comparative literature (Yu et al., 2019; Li et al., 2020; Luo et al., 2021; Wang et al., 2023), our methods are trained for a fixed number of 6,000 iterations to obtain the final model. Additionally, our models all use a batch size of 4, with a labeled data quantity of 2. We tested the performance of the models, and all experiments were conducted on NVIDIA® GeForce 3090 24GB running Ubuntu 20.04 and PyTorch 1.11.0.

## 4.2 DATASETS AND METRICS

In the experiment, we selected two datasets with different modalities and utilized four distinct metrics to assess the performance of the model. For each dataset, 80% of the data was used as the training set and 20% as the test set.

**LA Dataset.** It (Xiong et al., 2021) includes 100 3D gadolinium-enhanced MR imaging volumes of left atrial with an isotropic resolution of $0.625 \times 0.625 \times 0.625 mm^3$ and the corresponding ground truth labels. For pre-processing, we first normalize all volumes to zero mean and unit variance, then crop each 3D MRI volume with enlarged margins according to the targets.

**Pancreas-NIH Dataset.** It (Roth et al., 2015) provides 82 contrast-enhanced abdominal 3D CT volumes of pancreas with manual annotation. The size of each CT volume is $512 \times 512 \times D$, where $D \in [181, 466]$. In pre-processing, we use the soft tissue CT window of $[-120, 240]$ HU, crop the CT scans centering at the pancreas region, and enlarge margins with 25 voxels.

**Metrics.** Following (Wang et al., 2023; Yu et al., 2019; Luo et al., 2021; Xu et al., 2022; Bai et al., 2023; Li et al., 2020), we use four metrics to evaluate model performance, including regional sensitive metrics: Dice similarity coefficient (Dice) (Yu et al., 2019), Jaccard similarity coefficient (Jaccard) (Luo et al., 2021), and edge sensitive metrics: 95% Hausdorff Distance (95HD) (Xu et al., 2022) and Average Surface Distance (ASD) (Bai et al., 2023).

## 4.3 ABLATION STUDY

We evaluated the rationality of the DDM design on the LA dataset. For fairness, we trained our model on the LA dataset using 10% of the labeled data, employing VNet and the classic Mean Teacher architecture.

For the choice of guidance methods, we compared the use of no guidance (fixed alpha) with Teacher-guided and Student-guided methods. The experimental results are shown in Table 1. The results indicate that the Student-guided method achieved a 2.11% improvement in Dice, a 3.14% improvement in Jaccard, a reduction of 1.95 in 95HD, and a reduction of 0.19 in ASD compared to the Teacher-guided method. Moreover, the Teacher-guided method also showed performance improvements over the fixed alpha approach. This demonstrates that dynamically updating the EMA decay coefficient can better select the alpha value and enhance the representation capability of the Mean Teacher. Furthermore, choosing Student-guided guidance can further optimize the selection of alpha and improve the representation capability of the student model.

Table 1: Ablation results about **Student-Guided** on LA dataset

| Method | Labeled | Metrics | | | |
|---|---|---|---|---|---|
| | | Dice↑ | Jaccard↑ | 95HD↓ | ASD↓ |
| Mean Fixed | 8(10%) | 81.70 | 70.77 | 10.36 | 2.76 |
| Teacher-Guided | 8(10%) | 83.30 | 71.76 | 10.13 | 2.47 |
| Student-Guided | 8(10%) | **85.59** | **74.90** | **8.18** | **2.28** |

## 4.4 COMPARISON WITH OTHER METHODS

We compared our approach with previous state-of-the-art methods on LA dataset and Pancreas-NIH dataset. We chose VNet as baseline models for comparison. For the selected alternative models, we opted for UA-MT (Yu et al., 2019) with uncertainty estimation, BCP (Bai et al., 2023) using bidirectional CutMix (Yun et al., 2019), MCF (Wang et al., 2023) with model-level regularization, and PMT (Gao et al., 2024) and AD-MT (Zhao et al., 2023) using temporary policy and mean teacher framework, being state-of-the-art results.

**Comparison on LA Dataset.** We conducted a cross-model comparison on the classic LA dataset. We tested with 5% and 10% of labeled data. Results of the experiments are presented in **Table 2**. To provide a more intuitive demonstration of the performance of various models on the LA dataset, we have selected some representative results for visualization, as illustrated in **Fig. 3**.

Our DDM was applied to methods using the MT framework, namely UA-MT, AD-MT, and PMT. Our approach brought significant performance improvements over the original methods and outper-

Table 2: Comparison results on LA dataset with 5% and 10% labeled data

| Method | Labeled | Dice↑ | Jaccard↑ | 95HD↓ | ASD↓ |
|--------|---------|-------|----------|-------|------|
| | | Metrics | | | |
| VNet | 4(5%) | 52.55 | 39.60 | 47.05 | 9.87 |
| UA-MT | 4(5%) | 82.26 | 70.98 | 13.71 | 3.82 |
| MCF | 4(5%) | - | - | - | - |
| BCP | 4(5%) | 88.02 | 78.72 | 7.90 | 2.15 |
| PMT$^{SOTA}$ | 4(5%) | 89.47 | 81.04 | 6.45 | 1.86 |
| AD-MT$^{SOTA}$ | 4(5%) | 89.63 | 81.28 | 6.56 | 1.85 |
| UA-MT+DDM | 4(5%) | 83.76(+1.50) | 72.53(+1.55) | 25.59(+11.88) | 7.15(+3.33) |
| PMT+DDM | 4(5%) | 89.83(+0.36) | 81.61(+0.57) | 6.48+0.03 | 1.81(-0.05) |
| AD-MT+DDM | 4(5%) | **90.54**(+0.91) | **82.86**(+1.58) | **6.13**(-0.43) | **1.66**(-0.19) |
| VNet | 8(10%) | 82.74 | 71.72 | 13.35 | 3.26 |
| UA-MT | 8(10%) | 86.28 | 76.11 | 18.71 | 4.63 |
| MCF | 8(10%) | 88.71 | 80.41 | 6.32 | 1.90 |
| BCP | 8(10%) | 89.62 | 81.31 | 6.81 | 1.76 |
| PMT$^{SOTA}$ | 8(10%) | 90.81 | 83.23 | 5.61 | 1.50 |
| AD-MT$^{SOTA}$ | 8(10%) | 90.55 | 82.79 | 5.81 | 1.70 |
| UA-MT+DDM | 8(10%) | 86.74(+0.46) | 76.98(+0.87) | 14.69(-4.02) | 3.95(-0.68) |
| PMT+DDM | 8(10%) | 91.02(+0.21) | 83.57(+0.34) | **5.28**(-0.33) | 1.53(+0.03) |
| AD-MT+DDM | 8(10%) | **91.34**(+0.79) | **84.14**(+1.35) | 6.28(+0.47) | **1.39**(-0.31) |

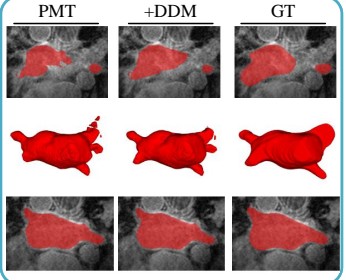 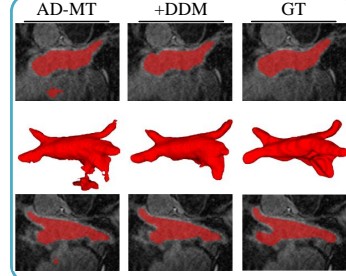 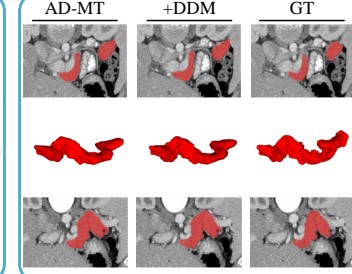

Figure 3: 2D & 3D segmentation visualization of improvement of DDM against MT frameworks under 10% labeled on LA (left and middle) and 20% pancreas (right) dataset.

formed previous models across all four metrics with both 5% and 10% of supervised data. With 5% of the data, compared to the best results from previous work, our ADMT+DDM improved Dice by 0.91%, Jaccard by 1.58%, reduced 95HD by 0.32, and reduced ASD by 0.19. With 10% of the data, our ADMT+DDM improved Dice by 0.53%, Jaccard by 0.91%, and reduced ASD by 0.11. Compared to state-of-the-art results, our model enhanced SOTA methods and achieved leading performance on most metrics using only half the data. For instance, with just 5% labeled data, we surpassed the performance of AD-MT using 10% labeled data. These results indicate that our DDM module can enhance performance across different proportions of labeled data and various MT framework models, achieving excellent results in the left atrium segmentation task. This demonstrates that DDM, as a plug-and-play module, can fully leverage the potential of the MT framework.

**Comparison on Pancreas-NIH Dataset.** We conducted a cross-model comparison on the classic Pancreas dataset. Detailed results of the experiments are presented in **Table 3**. We tested with 10% and 20% of labeled data. To provide a more intuitive demonstration of the performance of various models on the Pancreas-NIH dataset, we have selected some representative results for visualization, as illustrated in **Fig. 3**. Areas with inaccurate segmentation have been annotated accordingly. It is worth noting that the results in the table clearly indicate that Pancreas-NIH dataset is significantly more challenging than LA dataset.

Table 3: Comparison results on Pancreas-NIH dataset with 10% and 20% labeled data

| Method | Labeled | Metrics | | | |
|---|---|---|---|---|---|
| | | Dice↑ | Jaccard↑ | 95HD↓ | ASD↓ |
| VNet | 6(10%) | 55.60 | 41.74 | 45.33 | 18.63 |
| UA-MT | 6(10%) | 66.34 | 53.21 | 17.21 | 4.57 |
| BCP | 6(10%) | 73.83 | 59.24 | 12.71 | 3.72 |
| MCF | 6(10%) | - | - | - | - |
| PMTSOTA | 6(10%) | 81.00 | 68.33 | 6.36 | 1.62 |
| AD-MTSOTA | 6(10%) | 80.21 | 67.51 | 7.18 | 1.66 |
| UA-MT+DDM | 6(10%) | 67.03(+0.69) | 53.60(+0.39) | 24.05(+6.84) | 8.00(+3.43) |
| PMT+DDM | 6(10%) | **82.28**(+1.28) | **70.05**(+1.72) | **5.70**(-0.66) | **1.54**(-0.08) |
| AD-MT+DDM | 6(10%) | 81.45(+1.24) | 68.99(+1.48) | 6.00(-1.18) | 1.56(-0.10) |
| VNet | 12(20%) | 72.38 | 58.26 | 19.35 | 5.89 |
| UA-MT | 12(20%) | 65.25 | 50.83 | 27.17 | 9.17 |
| BCP | 12(20%) | 82.91 | 70.97 | 6.43 | 2.25 |
| MCF | 12(20%) | 75.00 | 61.27 | 11.59 | 3.27 |
| PMTSOTA | 12(20%) | 83.22 | 71.52 | 7.60 | 1.89 |
| AD-MTSOTA | 12(20%) | 82.61 | 70.70 | **4.94** | 1.38 |
| UA-MT+DDM | 12(20%) | 67.69(+2.44) | 53.05(+1.22) | 23.03(-4.14) | 8.45(-0.72) |
| PMT+DDM | 12(20%) | 83.28(+0.06) | 71.60(+0.08) | 8.50(+0.90) | 2.18(+0.29) |
| AD-MT+DDM | 12(20%) | **83.29**(+0.68) | **71.67**(+0.97) | 5.05(+0.11) | **1.26**(-0.12) |

We also integrated the DDM into models using the MT framework. The results show that models with the DDM outperform previous models across all four metrics with 10% and 20% labeled data. With 10% of the data, compared to the best results from previous work, our PMT+DDM improved Dice by 1.28%, Jaccard by 1.72%, reduced 95HD by 0.66, and reduced ASD by 0.08. With 20% of the data, our ADMT+DDM also improved Dice by 0.07%, Jaccard by 0.15%, and reduced ASD by 0.12. These results demonstrate that our model achieves excellent results in the pancreas segmentation task with different proportions of labeled data.

# 5 CONCLUSION

In this paper, we propose a semi-supervised medical image segmentation framework named Meta Mean Teacher. Meta Mean Teacher introduces the DDM module to dynamically adjust the EMA decay coefficient, a critical hyperparameter in the Mean Teacher model. We demonstrate the effectiveness of the DDM model through simple theoretical derivations and experiments. Compared to a more straightforward approach, Teacher-guided, our Student-guided selection enhances the model's representation capability. We argue that it is challenging for the Teacher model to fit the segmentation distribution, whereas it is simpler for the Teacher to learn how to provide pseudo-labels that are more beneficial for the Student. In comparative experiments with other methods, the MT framework methods incorporating DDM achieved state-of-the-art accuracy, significantly surpassing previous methods and maintaining this advantage even with more limited data and more challenging tasks. This indicates that our proposed DDM not only enhances the representation capability of the classic Mean Teacher but also, as a plug-and-play component, has extensibility and can generalize to different models, demonstrating the versatility of our approach.

**Limitations and Future Work.** Despite the excellent performance of Meta Mean Teacher, the selection of a more reasonable guidance method remains to be explored. Even with Student-guided approach, the essence of the Mean Teacher still involves searching within a highly limited space, which restricts the potential of the Mean Teacher. Investigating more reasonable guidance methods, integrating them into the Meta Mean Teacher paradigm, and evaluating their potential to further enhance performance is a topic worthy of future research. Additionally, applying this work to methods beyond medical imaging to further explore the generalizability of Meta Mean Teacher is also a necessary area of study.

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

## A APPENDIX

You may include other additional sections here.

