# OpenReview forum: "Learning to Teach: Improving Mean Teacher in Semi-supervised Medical Image Segmentation with Dynamic Decay  Modulation"
_ICLR.cc/2025/Conference — Submitted to ICLR 2025_

### Official Review · Reviewer_AjMw · 2024-10-21

**Soundness:** 3
**Presentation:** 3
**Contribution:** 2
**Rating:** 5
**Confidence:** 4

**Summary:**

This paper explores the EMA decay coefficient within the MT semi-supervised framework, fully tapping into the potential of the MT framework. Additionally, it introduces a novel meta-learning strategy to dynamically find the optimal EMA decay coefficient during the training process. Experiments conducted on two medical image datasets demonstrate that this method achieves superior performance.

**Strengths:**

1. This paper introduces a novel meta-learning strategy to adjust the EMA decay coefficient, fully tapping into the potential of the MT semi-supervised framework.
2. This paper introduces a strategy to adjust the EMA decay coefficient to improve semi-supervised segmentation performance, which could be a meaningful contribution to this field.
3. The extensive experimental results show the effectiveness of the proposed method.

**Weaknesses:**

1. I have not observed many innovative aspects in the application of meta-learning to the field of semi-supervised medical image segmentation. Part of the reason for this is the clarity of the writing; it is currently unclear what significant differences exist between the proposed DDM and previous meta-learning strategies. If there are no substantial differences, then the methodological contribution of this approach appears to be quite limited.
2. Could the authors explain what potential drawbacks a fixed EMA decay coefficient might have on the MT framework, particularly in the context of medical image processing?
3. The motivation is unclear. I do not understand why a dynamic change in $\alpha$ would have a greater advantage compared to a fixed value. $\alpha$ can be understood as the weight distribution between the teacher model’s parameters and the student model’s parameters during the iterative update process, with the teacher model’s weight being overwhelmingly dominant. I question the assumption that dynamically varying $\alpha$ between 0.95 and 0.99 is necessarily better than a fixed value of 0.97. Could you provide a plot showing how $\alpha$ changes dynamically over training iterations in the experiments?
4. In Equation 2, what criteria does DDM use to derive αm? Is there a relationship between $\alpha_m$ and these two losses? For example, if the teacher model has a lower loss, should $\alpha_m$ be larger? Please explain.
5. The notation in the paper is somewhat confusing. In Equation 1, what are the differences between $\Theta_s^*$ and $\Theta_s$, and between $\Theta_s$ and $\theta_s$? Additionally, what is meant by meta data $\mathcal{D}_m$, and how does it differ from labeled data and unlabeled data? Furthermore, the $\mathcal{L}_m$ formula is missing; I suggest adding it.
6. What is the initial value of $\alpha$? Has an ablation study been conducted to verify the impact of $\alpha$?
7. The authors mention that DDM can be promoted as a plug-and-play component for different models. However, I think DDM has limitations. For instance, how would DDM be applied to commonly used pseudo-labeling methods based on CPS (Semi-Supervised Semantic Segmentation with Cross Pseudo Supervision)?

**Questions:**

Please refer to the weakness.

---

### Official Review · Reviewer_91z3 · 2024-10-29

**Soundness:** 2
**Presentation:** 2
**Contribution:** 2
**Rating:** 3
**Confidence:** 4

**Summary:**

The paper introduces the Meta Mean Teacher framework, a novel approach to improve semi-supervised medical image segmentation. Traditional Mean Teacher models use a fixed Exponential Moving Average (EMA) decay coefficient to update the teacher model, but this fixed value often limits model effectiveness. Meta Mean Teacher addresses this limitation by introducing a Dynamic Decay Modulation (DDM) module that adaptively adjusts the EMA decay coefficient based on training dynamics. This dynamic adjustment optimizes the student-teacher learning process, enabling better performance in tasks with limited labeled data.

Key contributions of this work include:

1. Adaptive EMA Decay: The DDM module optimizes the EMA decay coefficient, enhancing the model's adaptability and enabling it to capture richer training representations.
2. Plug-and-Play Architecture: Meta Mean Teacher is designed to integrate seamlessly into existing models, improving performance across various Mean Teacher-based methods.

**Strengths:**

1. Dynamic Adaptability: By incorporating the Dynamic Decay Modulation (DDM) module, the framework dynamically adjusts the EMA decay coefficient (α) during training. This adaptability ensures that the teacher model evolves effectively with the student model, allowing more precise guidance as training progresses. This approach addresses a common limitation in fixed-coefficient Mean Teacher models, which often fail to account for varying training dynamics.

2. Plug-and-Play Module: The Meta Mean Teacher framework is designed as a modular system, making it highly compatible with existing models based on the Mean Teacher architecture. This modularity allows easy integration into various semi-supervised frameworks like UA-MT, AD-MT, and PMT.

3. Enhanced Stability and Robustness: The framework benefits from the Mean Teacher method’s inherent stability due to EMA but improves upon it by learning an optimal decay coefficient through meta-learning techniques.

**Weaknesses:**

1. High Computational Overhead: The adaptive EMA adjustment via DDM introduces complexity and requires more computational resources. The dynamic adjustment process, which includes cloning and iterative updates of both teacher and student models, may not be feasible for real-time or resource-limited applications, especially when processing large 3D medical imaging data.

2. Limited Exploration of Other Adaptive Techniques: While the paper focuses on dynamically adjusting the EMA decay coefficient, other hyperparameters (like learning rates or loss weight factors) could also impact model performance in semi-supervised learning. The focus on only one parameter might restrict the overall optimization potential, as additional adjustments could further enhance the segmentation quality.

**Questions:**

1. On the impact of α=0.01: Why does the model show improvement when α is set to 0.01? This result seems to contradict the explanation provided in Section 3.1.

2. The use of fixed α in the ablation experiments in Section 4.3: In Section 4.3, why was the average fixed α method chosen for comparison? From Figure 1, we can see that lower α values ​​(such as 0.03, 0.05, and 0.1) significantly degrade the performance. In contrast, α of 0.97 achieves performance higher than 0.84. Doesn't this general average comparison seem a bit biased?

3. The impact of α greater than 0.5: From Table 1, we can see that when α is greater than 0.5, its impact on performance becomes less significant. Test whether randomly selected α values ​​greater than 0.5 are beneficial, thereby potentially improving the results?

4. Suspicion about the data in Section 4.4 compared with other methods: Why is your experimental setting different from that in "Alternative Diversified Teaching of Semi-Supervised Medical Image Segmentation", but most of the state-of-the-art data (including LA and Pancreas-NIH datasets) are the same as the data in that paper? Does this indicate that the data is directly borrowed?

5. The m symbol in Figure 2: What does "m" mean in Figure 2? Is this symbol redundant?

---

### Official Review · Reviewer_mwbG · 2024-11-02

**Soundness:** 3
**Presentation:** 3
**Contribution:** 1
**Rating:** 3
**Confidence:** 4

**Summary:**

This paper presents the 'Meta Mean Teacher', an approach for semi-supervised medical image segmentation. Building on the Mean Teacher model, which leverages exponential moving average (EMA) to create a stable teacher model from a student model, this framework introduces the Dynamic Decay Modulation (DDM) module. DDM dynamically adjusts the EMA decay coefficient based on both the student and teacher losses, improving the model's adaptability during training.

**Strengths:**

The paper addresses semi-supervised learning in medical image segmentation with a novel meta-learning approach, introducing the Dynamic Decay Modulation (DDM) module to adjust the EMA decay coefficient dynamically.

The paper strengthens its empirical evaluation by testing on three datasets, covering different imaging modalities.

**Weaknesses:**

While the paper builds on the Mean Teacher model, which is well-established in semi-supervised learning, it may lack substantial novelty as the framework mainly modifies an existing approach. Although the Dynamic Decay Modulation (DDM) module adds a new layer of adaptability, many similar extensions to Mean Teacher already exist, potentially limiting the paper's contribution to novel methodology.


The experimental scope appears limited as it only includes limited number of baseline methods, i.e. Mean Teacher variations like UAMT with UNet and VNet, models that have already been well-explored in this context. The paper’s experiments may be restricted by a limited range of labeled-to-unlabeled data ratios, which does not fully capture the model’s performance across different semi-supervised settings. Testing with a wider variety of label-scarcity scenarios would offer more robust insights into the framework's adaptability and practical applicability in real-world cases where data availability varies.

**Questions:**

(1) How do you ensure that comparisons are fair in semi-supervised learning scenarios? For example, I understand that in some cases, we can control the percentage of labeled and unlabeled data, such as using 5% or 10% labeled data. However, the feature distribution of labeled and unlabeled data cannot be guaranteed to be the same.


(2) The exclusive use of VNet as the backbone may limit the generalizability of the results, as it does not reflect performance across more commonly used architectures like UNet or newer ViT-based UNets.

(3) In Table 2, I observe that VNet’s performance is significantly lower than others when only 5% of data is labeled, but it is only slightly lower when 10% is labeled. Could you explain why this discrepancy occurs? Additionally, could you provide more results for cases with 20%, 50%, 80%, and 90% labeled data, if available?

(4) In table 3, why VNet outperforms UA-MT when 20% are labeled.

---

### Meta-Review · Area_Chair_owf3 · 2024-12-19

**Metareview:**

This paper introduces the Meta Mean Teacher, a framework for semi-supervised medical image segmentation that improves upon traditional Mean Teacher models by incorporating a Dynamic Decay Modulation (DDM) module, which adaptively adjusts the EMA decay coefficient based on training dynamics, resulting in enhanced adaptability and performance in tasks with limited labeled data.

Reviewers found that the strengths of this paper lie in its novel Dynamic Decay Modulation module, which dynamically adjusts the EMA decay coefficient during training, addressing limitations of traditional Mean Teacher models and improving adaptability, stability, and performance. The framework’s modular design allows seamless integration into existing semi-supervised architectures.  On the other hand, the main weaknesses of the paper include limited novelty, unclear motivation, and restricted experimental scope, as noted by multiple reviewers. The Dynamic Decay Modulation module lacks significant innovation compared to existing meta-learning strategies, and its benefits over a fixed EMA decay coefficient remain insufficiently justified (Reviewers mwbG, AjMw). The experimental evaluation is limited, with a narrow range of baseline methods and labeled-to-unlabeled data ratios, restricting insights into real-world applicability (Reviewer mwbG). Additionally, the high computational overhead of the DDM module may hinder its feasibility for resource-constrained or real-time applications (Reviewer 91z3), and the focus on a single hyperparameter (EMA decay) overlooks opportunities for optimizing others (Reviewer 91z3).

No rebuttal was submitted.

All three reviewers leaned towards rejection. After carefully considering the reviewers' comments and the lack of a rebuttal from the authors, I have decided to reject this paper. While the Dynamic Decay Modulation (DDM) module introduces an interesting extension to the Mean Teacher framework, reviewers noted significant concerns regarding the limited novelty, unclear motivation, and restricted experimental scope. The insufficient exploration of alternative adaptive techniques, high computational overhead, and lack of ablation studies further weaken the paper’s contributions. Without a rebuttal to address these critical issues, the paper does not meet the standard required for acceptance.

**Additional Comments On Reviewer Discussion:**

The main weaknesses of the paper include limited novelty, unclear motivation, and restricted experimental scope, as noted by multiple reviewers. The Dynamic Decay Modulation module lacks significant innovation compared to existing meta-learning strategies, and its benefits over a fixed EMA decay coefficient remain insufficiently justified (Reviewers mwbG, AjMw). The experimental evaluation is limited, with a narrow range of baseline methods and labeled-to-unlabeled data ratios, restricting insights into real-world applicability (Reviewer mwbG). Additionally, the high computational overhead of the DDM module may hinder its feasibility for resource-constrained or real-time applications (Reviewer 91z3), and the focus on a single hyperparameter (EMA decay) overlooks opportunities for optimizing others (Reviewer 91z3).

No rebuttal was submitted.

---

### Decision · Program_Chairs · 2025-01-22

Reject